# Emphysema Quantification Using Ultra-Low-Dose Chest CT: Efficacy of Deep Learning-Based Image Reconstruction

**DOI:** 10.3390/medicina58070939

**Published:** 2022-07-15

**Authors:** Jeong-A Yeom, Ki-Uk Kim, Minhee Hwang, Ji-Won Lee, Kun-Il Kim, You-Seon Song, In-Sook Lee, Yeon-Joo Jeong

**Affiliations:** 1Department of Radiology, Pusan National University Yangsan Hospital, Pusan National University School of Medicine, Yangsan 50612, Korea; mdyja@naver.com (J.-A.Y.); kikim@pusan.ac.kr (K.-I.K.); 2Department of Internal Medicine and Biomedical Research Institute, Pusan National University Hospital, Busan 49241, Korea; uk303@hanmail.net; 3Department of Radiology and Biomedical Research Institute, Pusan National University Hospital, Pusan National University School of Medicine, Busan 49241, Korea; hmh8807@naver.com (M.H.); monophobia00@hanmail.net (J.-W.L.); yssongrad@gmail.com (Y.-S.S.); lis@pusan.ac.kr (I.-S.L.)

**Keywords:** emphysema, low dose CT, quantitative analysis, deep learning

## Abstract

*Background and Objectives*: Although reducing the radiation dose level is important during diagnostic computed tomography (CT) applications, effective image quality enhancement strategies are crucial to compensate for the degradation that is caused by a dose reduction. We performed this prospective study to quantify emphysema on ultra-low-dose CT images that were reconstructed using deep learning-based image reconstruction (DLIR) algorithms, and compared and evaluated the accuracies of DLIR algorithms versus standard-dose CT. *Materials and Methods*: A total of 32 patients were prospectively enrolled, and all underwent standard-dose and ultra-low-dose (120 kVp; CTDIvol < 0.7 mGy) chest CT scans at the same time in a single examination. A total of six image datasets (filtered back projection (FBP) for standard-dose CT, and FBP, adaptive statistical iterative reconstruction (ASIR-V) 50%, DLIR-low, DLIR-medium, DLIR-high for ultra-low-dose CT) were reconstructed for each patient. Image noise values, emphysema indices, total lung volumes, and mean lung attenuations were measured in the six image datasets and compared (one-way repeated measures ANOVA). *Results*: The mean effective doses for standard-dose and ultra-low-dose CT scans were 3.43 ± 0.57 mSv and 0.39 ± 0.03 mSv, respectively (*p* < 0.001). The total lung volume and mean lung attenuation of five image datasets of ultra-low-dose CT scans, emphysema indices of ultra-low-dose CT scans reconstructed using ASIR-V 50 or DLIR-low, and the image noise of ultra-low-dose CT scans that were reconstructed using DLIR-low were not different from those of standard-dose CT scans. *Conclusions*: Ultra-low-dose CT images that were reconstructed using DLIR-low were found to be useful for emphysema quantification at a radiation dose of only 11% of that required for standard-dose CT.

## 1. Introduction

Pulmonary emphysema is a major disease that encompasses chronic obstructive pulmonary disease (COPD) along with chronic bronchitis. Smoking is the leading cause of COPD, and as the smoking rate increases, the prevalence and mortality of COPD is emerging as a major problem in public health along with the increase in lung cancer [1]. The pulmonary function test (PFT) is the most basic test that is used to diagnose and classify the severity classification of COPD [1], but its sensitivity for the diagnosis of mild disease is low [2]. On the other hand, computed tomography (CT) is more accurate and sensitive for diagnosing emphysema and evaluating its severity and extent as it well depicts morphological changes in the lung parenchyma well [3]. In addition, CT has been reported to be excellent at quantifying emphysema based on measurements of low attenuation areas, either the percentage of voxels below −950 Hounsfield Unit (HU) or the 15th percentile of the attenuation curve [4,5,6,7,8,9].

Recently, low-dose CT has been widely used for the early detection of lung cancer, and as the majority of the subjects that are involved are smokers, interest in quantifying emphysema by low-dose CT is also increasing. Although the emphysema quantification is affected by the CT acquisition parameters such as radiation dose, slice thickness, and reconstruction filters [10,11], when emphysema quantification using low-dose CT was performed, the mean lung attenuation was found to be similar to that which was obtained using standard-dose CT, and emphysema indices that were calculated using volume ratios of the low attenuation areas concurred [12].

Although low-dose CT only requires a radiation dose of 1~2 mSv, the risks that are posed by radiation exposure due to regular follow-up examinations cannot be ignored. On the other hand, ultra-low-dose CT requires only sub-millisievert exposures (0.1~0.3 mSv), which are similar to those that are required for chest radiography. However, when ultra-low-dose CT images are reconstructed using conventional methods, image qualities are degraded by high-quantum noise to an extent that prevents their use in daily practice [13].

The recently developed and commercialized deep learning-based image reconstruction (DLIR) techniques were trained using high-quality filtered back projection (FBP) images, validated iterative reconstruction (IR) images, or denoised images from conventional IR and FBP images [14,15]. The DLIR method using high-quality FBP images is trained to differentiate noise from signals and to suppress the effects of noise without impacting anatomical and pathological structures [15]. Therefore, ultra-low-dose CT images that are reconstructed using a DLIR method can have the similar quality as standard-dose CT images. The purpose of this study is to quantify emphysema on ultra-low-dose CT images that were reconstructed using DLIR, and to compare and evaluate the accuracies of DLIR algorithms versus standard-dose CT.

## 2. Materials and Methods

### 2.1. Study Design and Subjects

The Institutional review board of Pusan National University Hospital approved this prospective study (H-2001-013-087; approval date, 31 January 2020), which involved the standard-dose and ultra-low-dose CT imaging of a cohort of patients during one examination. All the study subjects had given informed consent for the study design and the use of CT data for research. We performed this prospective study during the period from February 2020 to September 2020.

The inclusion criteria of this study were as follows: (1) Clinically diagnosed with COPD, and (2) adult with more than 5% emphysema confirmed on a previous chest CT scan. The exclusion criteria were as follows: (1) Presence of severe artifacts that interfere with CT quantification, (2) presence of a large proportion of atelectasis and pulmonary fibrosis on the previous CT scan, and (3) patients with a history of lung resection. The study population consisted of 32 patients (all men; mean (±SD) age, 71.28 ± 5.96 years; age range, 60–84 years) with a history of COPD. Of these study subjects, 14 were current smokers and 18 were ex-smokers, and they had a mean smoking history of 50.22 ± 24.94 (range, 5–120) pack-years. All underwent PFTs and CT examinations within 3 days of each other. Their median modified medical research council dyspnea scale (mMRC. Appendix A) was 1 (range, 0–2) and their mean COPD assessment test (CAT) score (Appendix A) was 5.75 ± 4.98 (range, 0–25). A total of three patients had both mMRC of ≥2 and a CAT score of ≥10. Body-mass indices (BMI) of all the patients were measured at the time of CT examination. A total of fifteen patients were BMI ≤ 23 kg/m^2^ (21.1 ± 1.5; range, 17.8~23.0) and 17 were BMI > 23 kg/m^2^ (24.9 ± 1.9; range, 23.1~31.1).

### 2.2. CT Data Acquisition and Reconstruction

The 32 subjects underwent standard-dose and ultra-low-dose CT scans of the whole chest without contrast material in one single examination using a 256-slice multidetector CT (Revolution CT; GE Healthcare, Waukesha, WI, USA) using the following parameters: tube current, 60–120 mAs with automatic exposure control for standard-dose CT and 10 mAs for ultra-low-dose CT; tube potential, 120 kVp; volume CT dose index, ≤5 mGy (for standard-dose CT) or <0.7 mGy (for ultra-low-dose CT); collimation, 128 × 0.625 mm detector configuration; rotation time, 0.5 s. All the CT scans were performed in the supine position during a deep inspiratory breath-hold.

Standard-dose CT images were reconstructed using FBP with a standard kernel. For ultra-low-dose CT, the images were reconstructed using FBP with a standard kernel, an adaptive statistical iterative reconstruction (ASIR-V; GE Healthcare Technologies, Waukesha, WI, USA) and a commercially available DLIR (TrueFidelity; GE Healthcare Technologies, Waukesha, WI, USA). For ASIR-V, 50% blending with FBP (ASIR-V 50) was used, and for DLIR, all of the three selectable reconstruction strength levels (low, medium, and high) were used to control the amount of noise reduction. A total of six image sets (FBP for standard-dose CT, FBP, ASIR-V 50, DLIR-low, DLIR-medium, and DLIR-high for ultra-low-dose CT) were obtained for each patient.

### 2.3. Image Analysis

To assess the objective image qualities, image noise was assessed using standard deviation (SD) of tracheal lumen attenuation by drawing regions of interest inside the trachea in each image dataset. Quantitative analysis measurements of emphysema were obtained using an automated lung image analysis tool (Thoracic VCAR; GE Healthcare, Waukesha, WI, USA) that was installed on an image processing workstation (AW server version 3.1; GE Healthcare, Waukesha, WI, USA) for each image dataset. The total lung volumes (TLV), mean lung attenuations, and emphysema indices that were represented by relative area under threshold of −950 HU were obtained.

### 2.4. Statistical Analysis

Statistical analysis was performed using SPSS (version 20; IBM Corp., Armonk, NY, USA) and *p*-values of less than 0.05 were considered significant. The results for continuous data were expressed as means ± SDs. The paired Student t-test was used to compare the radiation doses that were administered during standard-dose and ultra-low-dose CT acquisitions. One-way repeated measures ANOVA was used to compare image noises and quantitative measurements of emphysema between six image datasets. Image noises and emphysema indices between six image datasets according to BMI (BMI ≤ 23 kg/m^2^ vs. BMI > 23 kg/m^2^) and the extent of emphysema (emphysema index ≤ 10% vs. >10%) were also compared using one-way repeated measures ANOVA. Agreement between quantitative emphysema measurements that were obtained by standard-dose CT and those that were obtained for the five ultra-low-dose datasets were assessed using Spearman correlation coefficients, and the mean measurement bias was calculated by Bland–Altman analysis.

## 3. Results

### 3.1. Radiation Dose

The mean dose length products were 244.92 ± 40.73 mGy cm for standard-dose CT and 27.54 ± 1.88 mGy cm for ultra-low-dose CT, and the mean effective doses were 3.43 ± 0.57 and 0.39 ± 0.03 mSv, respectively (*p* < 0.001).

### 3.2. Quantitative Measurements of Standard-Dose and Ultra-Low-Dose CT

Table 1 summarizes image noises, emphysema indices, total lung volumes, and the mean lung attenuations of standard-dose and ultra-low-dose CT. When standard-dose CT was used as the reference standard, the patients had a mean total lung volume of 5.46 ± 1.09 L, mean lung attenuation of −851.05 ± 24.44 HU, and mean emphysema index of 14.26 ± 11.02%. The total lung volume and mean lung attenuation of the five image datasets of ultra-low-dose CT scans were similar to those of standard-dose CT. Regardless of BMI or emphysema extent, emphysema indices of ultra-low-dose CT images that were reconstructed using ASIR-V 50 or DLIR-L were also similar to those of standard-dose CT, and the mean image noise of ultra-low-dose CT images that were reconstructed using DLIR-L was similar to that of standard-dose CT images (all *p*s > 0.05) (Figure 1a–f).

Quantitative measurements that were obtained using ultra-low-dose CT images were strongly correlated with corresponding standard-dose CT images, except for image noise (Table 2), for which ultra-low-dose CT images that were reconstructed using DLIR-H were moderately correlated with standard-dose CT images (*r* = 0.406, *p* = 0.021).

Bland–Altman analysis revealed that no relevant bias was observed for the total lung volume or the mean lung attenuation as determined using the five image datasets of ultra-low-dose CT scans (Table 3). With regards to the emphysema indices, no relevant bias was observed on ultra-low-dose CT scans reconstructed using ASIR-V 50 (mean overestimation, 0.55%) or DLIR-L (mean underestimation, 0.64%) (Figure 2 and Figure 3). For image noise, no relevant bias was observed on ultra-low-dose CT scans except on ultra-low-dose CT scans that were reconstructed using FBP. The mean measurement bias was lowest for ultra-low-dose CT scans that were reconstructed using DLIR-L (mean overestimation, 1.82 HU) (Figure 2 and Figure 3).

## 4. Discussion

Several studies have addressed the usefulness of low-dose or ultra-low-dose CT scans for emphysema quantification, but a few reports have been issued on the emphysema quantification using ultra-low-dose CT scans that are reconstructed with DLIR [16,17,18,19,20]. Our study showed that the total lung volume and mean lung attenuation of five image datasets of ultra-low-dose CT scans, emphysema indices of ultra-low-dose CT scans that were reconstructed using ASIR-V 50 or DLIR-low, and image noise of ultra-low-dose CT scans that were reconstructed using DLIR-low were not different from those of standard-dose CT scans.

It is well known that emphysema quantifications using CT, especially emphysema index, are well associated with both lung function test results and pathologic findings [4,5,6,7,8,9]. However, the risks that are associated with the exposure of patients to ionizing radiation must be considered prior to CT-based emphysema quantification. Low-dose CT offers a means of reducing the risks that are involved, but can cause problems because its use increases image noise. Several studies have attempted to reduce image noise by using iterative reconstruction when low-dose or ultra-low-dose CT is used for emphysema quantification [17,21]. Choo et al. evaluated the effects of iterative reconstruction on the quantitative analysis of lung parenchyma and airway measurements that were obtained using low-dose CT images and found that emphysema indices were decreased by using of iterative reconstruction [21]. Wang et al. evaluated the effects of iterative reconstruction on emphysema quantification that were performed using ultra-low-dose CT [17]. The absolute overestimation of the emphysema index was 2% on ultra-low-dose CT images that were reconstructed by iterative reconstruction, and 7% on ultra-low-dose CT images that were reconstructed using FBP versus low-dose CT images. Iterative reconstruction techniques synthesize projections by modelling the data collection process based on the noise properties of the imaged objects to allow dose reductions of 32~65% without increasing noise in the reconstructed images that are produced. Recently, one study evaluated the performance of various deep learning-based algorithms for emphysema quantification using a dataset with different low dose CT protocols and showed that intraclass correlation coefficients of emphysema index between standard-dose CT and converted low-dose CT scans using deep learning-based algorithms ranged from 0.85 to 0.94 [22]. Although deep learning-based algorithms can improve emphysema quantification from low-dose CT with heterogenous CT protocols, it is not clear whether this also applies to emphysema quantification for ultra-low-dose CT scans.

In the present study, emphysema indices of ultra-low-dose CT scans that were reconstructed using ASIR-V 50 or DLIR-L were similar to those of standard-dose CT scans. Image noise levels of ultra-low-dose CT images that were reconstructed using DLIR-L were similar to those that were obtained by standard-dose CT. The DLIR methods that were used in this study were trained using high-quality FBP datasets to learn how to differentiate noise and signals, and to suppress noise intelligently without impacting anatomical or pathological structures. The mean effective dose of ultra-low-dose CT was 0.39 ± 0.03 mSv, which was only about 11% of the mean effective dose of standard-dose CT. Therefore, DLIR was found to enable radiation dose reduction while preserving the image quality of the FBP technique.

The study has several limitations. First, it was performed in a small number of Asian men with emphysema, and it may be that the performance of ultra-low-dose CT for emphysema quantification is inferior in more diverse populations. Second, most of the patients that were included had mild symptoms (CAT score < 10 or mMRC < 2) and were at low risk (GOLD ≤ 2) of severe CODP. Further larger-scale studies on patients with a wider range of clinical statuses of COPD are required to confirm our results.

## 5. Conclusions

In conclusion, emphysema indices of ultra-low-dose CT scans that were reconstructed using ASIR-V 50 or DLIR-low, and image noise of ultra-low-dose CT scans that were reconstructed using DLIR-low were not different from those of standard-dose CT scans. Ultra-low-dose CT images that were reconstructed using DLIR-low were found to be useful for emphysema quantification at a radiation dose of only 11% of that required for standard-dose CT.

## Figures and Tables

**Figure 1 medicina-58-00939-f001:**
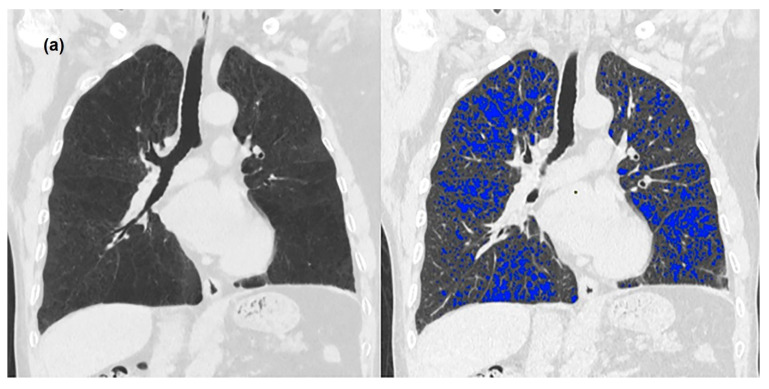
A 67-year-old man with chronic obstructive pulmonary disease. The patient had a 30-pack-year smoking history. Coronal CT and densitometric overlay images of standard-dose CT images reconstructed using filtered back projection (FBP) (**a**) and ultra-low-dose CT images that were reconstructed using FBP. (**b**) Adaptive statistical iterative reconstruction. (**c**) Deep learning-based image reconstruction-low-strength (DLIR-L). (**d**) DLIR- medium-strength (DLIR-M). (**e**) DLIR- high-strength (DLIR-H). (**f**) In densitometric overlay images, all voxels with a CT attenuation of <−950 HU are color-coded in blue. Emphysema indices of standard-dose CT, ultra-low-dose CT using FBP, ASIR-V 50, DLIR-L, DLIR-M, and DLIR-H are as follows; 12.63%, 19.65%, 16.29%, 14.88%, 12.78%, and 11.97%, respectively. Image noise levels of standard-dose CT, ultra-low-dose CT with FBP, ASIR-V 50, DLIR-L, DLIR-M, and DLIR-H are as follows; 19.6 HU, 31.9 HU, 25.9 HU, 18.7 HU, 15.7 HU, and 10.1 HU, respectively.

**Figure 2 medicina-58-00939-f002:**
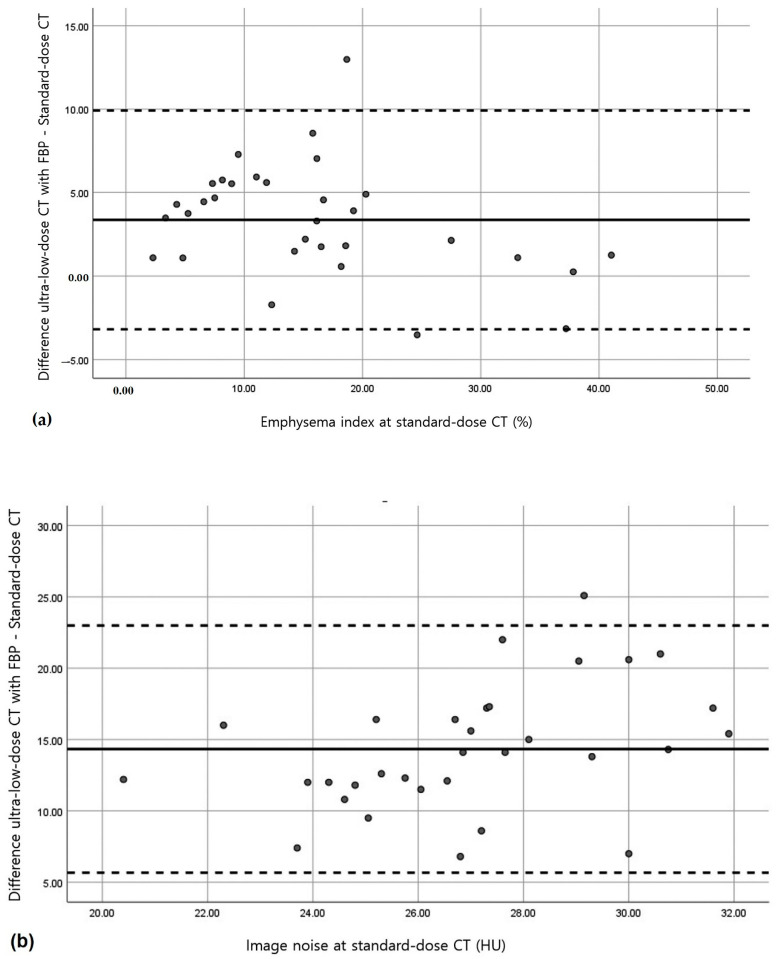
Bland–Altman analysis of standard-dose CT with reconstruction by filtered back projection (FBP) and of ultra-low-dose CT with FBP. (**a**,**b**) Plots show measurements of emphysema indices (**a**) and image noise values (**b**). The solid lines indicate the mean bias (overestimation or underestimation) for ultra-low-dose CT as compared with standard-dose CT. Dashed lines indicate 95% CIs.

**Figure 3 medicina-58-00939-f003:**
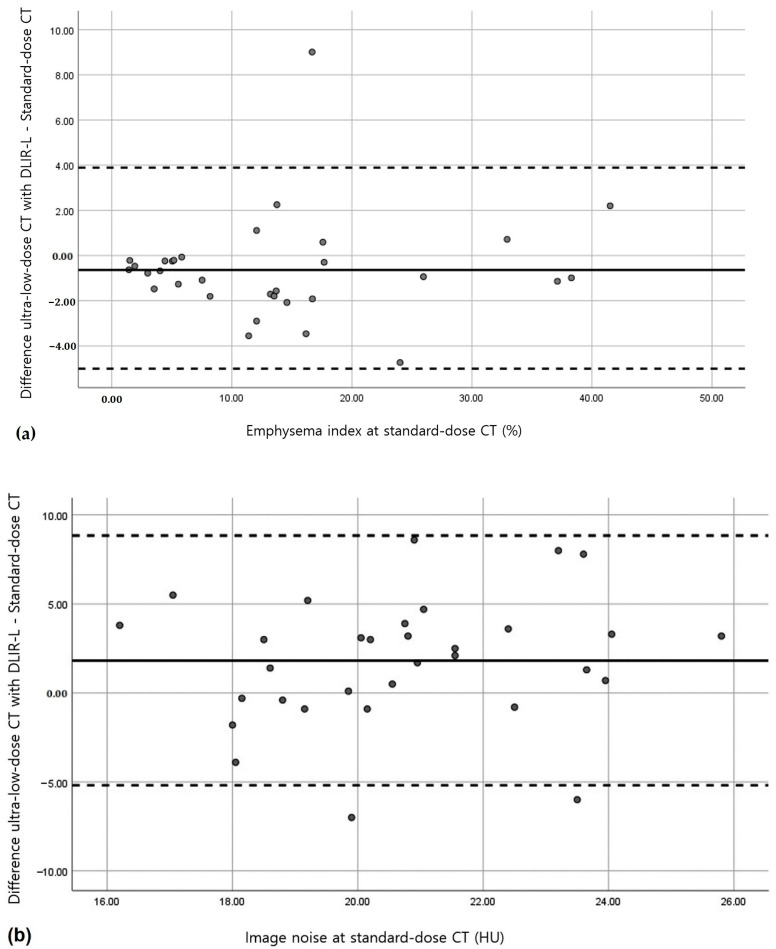
Bland–Altman analysis of standard-dose CT with reconstruction by filtered back projection (FBP) and ultra-low-dose CT using deep learning-based image reconstruction-low-strength (DLIR-L). (**a**,**b**) Plots show emphysema indices (**a**) and image noise values (**b**). The solid lines indicate the mean bias (overestimation or underestimation) for ultra-low-dose CT as compared with standard-dose CT. Dashed lines indicate 95% CIs.

**Table 1 medicina-58-00939-t001:** Quantitative measurements of emphysema and image noise in standard-dose and ultra-low-dose CT images.

Value	Standard-Dose CT	Ultra-Low-Dose CT					
	FBP	FBP	ASIR-V 50	DLIR-L	DLIR-M	DLIR-H	*p*
EI, %	14.26 (11.02)	17.62 (9.77)	14.81 (10.86)	13.62 (11.23)	12.29 (11.53)	11.76 (11.74)	<0.001
TLV, L	5.46 (1.09)	5.55 (1.13)	5.57 (1.16)	5.57 (1.16)	5.57 (1.16)	5.57 (1.16)	0.150
MLA, HU	−851.05 (24.44)	−849.41 (27.57)	−850.85 (27.67)	−851.25 (27.98)	−851.24 (27.88)	−850.58 (27.65)	0.380
Image noise, HU	19.80 (2.68)	34.13 (4.12)	25.38 (3.38)	21.62 (3.10)	16.27 (2.17)	10.47 (2.49)	<0.001

Data are presented as the means (standard deviations). ASIR-V 50, adaptive statistical iterative reconstruction; DLIR-L, deep learning-based image reconstruction-low-strength; DLIR-M, deep learning-based image reconstruction-medium-strength; DLIR-H, deep learning-based image reconstruction-high-strength; EI, emphysema index; FBP, filtered back projection; MLA, mean lung attenuation; HU, Hounsfield units; TLV, total lung volume; L, liter.

**Table 2 medicina-58-00939-t002:** Correlation coefficient of quantitative measurements and image noise between standard-dose CT and the five series of ultra-low-dose CT scans.

Value	Standard-Dose CT	Ultra-Low-Dose CT					
	FBP	FBP	ASIR-V 50	DLIR-L	DLIR-M	DLIR-H	*p*
EI	1	0.955	0.977	0.979	0.981	0.978	<0.001
TLV	1	0.925	0.933	0.932	0.933	0.934	<0.001
MLA	1	0.924	0.926	0.925	0.927	0.926	<0.001
Image noise	1	0.210	0.227	0.239	0.346	0.406	>0.05 *

The data shown are mean correlation coefficients. ASIR-V 50, adaptive statistical iterative reconstruction; DLIR-L, deep learning-based image reconstruction-low-strength; DLIR-M, deep learning-based image reconstruction-medium-strength; DLIR-H, deep learning-based image reconstruction-high-strength; EI, emphysema index; FBP, filtered back projection; MLA, mean lung attenuation; TLV, total lung volume. * ultra-low-dose CT images reconstructed with DLIR-H were moderately correlated with standard-dose CT images (*r* = 0.406, *p* = 0.021).

**Table 3 medicina-58-00939-t003:** The mean measurement bias between standard-dose CT and the five series of ultra-low-dose CT scans.

Value	Ultra-Low-Dose CT				
	FBP	ASIR-V 50	DLIR-L	DLIR-M	DLIR-H
Emphysema index	3.36 (3.35)	0.55 (2.33) *	−0.64 (2.31) *	−1.97 (2.24)	−2.50 (2.48)
Total lung volume	0.09 (0.43) *	0.11 (0.42) *	0.11 (0.42) *	0.12 (0.42) *	0.11 (0.41) *
Mean lung attenuation	1.65 (10.59) *	0.20 (10.49) *	−0.19 (10.74) *	−0.19 (10.55) *	0.48 (10.53) *
Image noise	14.33 (4.42)	5.58 (3.81) *	1.82 (3.58) *	−3.53 (2.80) *	−9.33 (2.82) *

Data are presented as the means (standard deviations). * No relevant bias is present. ASIR-V 50; adaptive statistical iterative reconstruction, DLIR-L; deep learning-based image reconstruction-low-strength, DLIR-M; deep learning-based image reconstruction-medium-strength, DLIR-H; deep learning-based image reconstruction-high-strength, FBP; filtered back projection.

## Data Availability

Data supporting the reported results may be provided on reasonable request.

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
