# Peer review of "Emphysema Quantification Using Ultra-Low-Dose Chest CT: Efficacy of Deep Learning-Based Image Reconstruction"

_medicina, 2022, doi:10.3390/medicina58070939_

Round 1

Reviewer 1 Report

The paper is well written, the ideas were clear exposed and easy to read. The paper described  clearly the useful of ultra high CT low dose in diagnostic of emphisema, in patients with COPD GOLD 2. For this patients put the diagnosis in earlier stage, could change the behavior and stop smoking/early treatment. 

May be in introduction more information, and more details in conclusion. More articles could be described in the introduction, and maybe the conclusion could be more explicit, not so succint; the conclusion are consistent with the evidence, but for the reader, could be more explicit.

Author Response

R1-1. The paper is well written, the ideas were clear exposed and easy to read. The paper described  clearly the useful of ultra-low dose CT in diagnostic of emphysema, in patients with COPD GOLD 2. For this patients put the diagnosis in earlier stage, could change the behavior and stop smoking/early treatment. 

-> Thank you for your comments.

R1-2. May be in introduction more information, and more details in conclusion. More articles could be described in the introduction, and maybe the conclusion could be more explicit, not so succint; the conclusion are consistent with the evidence, but for the reader, could be more explicit.

-> As you pointed out, we added more information and corresponding references to the Introduction for readers. In addition, we did not make the Conclusion implicitly, but expressed it consistent with the our study results in the revised version of manuscript.

Reviewer 2 Report

This study aimed to evaluated the differences among standard CT and ultra-low-dose CT in the assessment of emphysema index, lung volume, mean lung density using COPD patients, which can be of use in clinical settings. The detailed information would augment the significance of the findings.  

1.Can the subtypes of emphysema such as centrilobular emphysema or paraseptal emphysema affect the findings?

2. Can BMI affect the findings?

3. Can the extent or size of the emphysema affect the findings?

Author Response

R2-1.Can the subtypes of emphysema such as centrilobular emphysema or paraseptal emphysema affect the findings?

-> CT emphysema index is determined by automatic calculation from the CT data of the volume fraction of the lungs below -950 HU at full inspiration. Therefore, it may be affected by both radiation dose, reconstruction technique or emphysema extent, but not by subtype of emphysema.

R2-2. Can BMI affect the findings?

-> Thank you for your valuable comment. We re-analyzed whether BMI could influence the outcome as you pointed out.

R2-3. Can the extent or size of the emphysema affect the findings?

-> Thank you for your valuable comment. We re-analyzed whether emphysema extent could influence the outcome as you pointed out.

Reviewer 3 Report

Overall the article is well written. However, some issues must be resolved:

- Materials and Methods: how do the Authors justify the obviously increased dose of the double CT scan (standard-dose and ultra-low-dose) which every patient received? 

- Materials and Methods: please add the time interval during which the study was conducted.

- Materials and Methods: Inclusion Criteria of the study are missing; please add them to the manuscript section.

- Materials and Methods - line 78: please add a reference for the "median modified medical research council dyspnea scale" and for the "mean COPD assessment test (CAT) score".

- Materials and Methods - line 84: please rephrase " tube current, 60-120 mAs with automatic exposure control (for standard-dose CT) or 10 mAs (for ultra-low-dose CT);" to " tube current, 60-120 mAs with automatic exposure control for standard-dose CT and 10 mAs for ultra-low-dose CT;"

- Materials and Methods - line 106: please explain the abbreviation "HU" and add it to the text.

- the "p" in p-value should be written in italics lowercase ("p"). Please change it throughout the manuscript and in the tables.

- Materials and Methods: please explain what the "emphysema index" is, as it is only reported in the results (line 126) without explanation.

- Table 1 and 2: please re-write the table legends, as the punctuation is incorrect: the semicolon (";") is meant to link separate items and so should be used.

- Discussion: please add more recent studies to the discussion, also on the use of deep learning techniques.

Author Response

Overall the article is well written. However, some issues must be resolved:

R3-1. Materials and Methods: how do the Authors justify the obviously increased dose of the double CT scan (standard-dose and ultra-low-dose) which every patient received? 

-> As we mentioned it in the Materials and Methods, Institutional review board of our institution approved this prospective study, which involved the standard-dose and ultra-low-dose CT imaging of a cohort of patients during one examination because the radiation exposure of our study protocol (approximately 3.43mSv) was lower than the average radiation dose of the routine chest CT (approximately less than 6.1mSv). We fully explained this to the study subjects and obtained informed consents.

R3-2. Materials and Methods: please add the time interval during which the study was conducted.

-> Yes, we did.

R3-3. Materials and Methods: Inclusion Criteria of the study are missing; please add them to the manuscript section.

-> Yes, we did. We also added exclusion criteria.

R3-4. Materials and Methods - line 78: please add a reference for the "median modified medical research council dyspnea scale" and for the "mean COPD assessment test (CAT) score".

-> Yes, we did. We provided them in the Supplementary Tables.

R3-5. Materials and Methods - line 84: please rephrase " tube current, 60-120 mAs with automatic exposure control (for standard-dose CT) or 10 mAs (for ultra-low-dose CT);" to " tube current, 60-120 mAs with automatic exposure control for standard-dose CT and 10 mAs for ultra-low-dose CT;"

-> Thank you for pointing this out. We revised the sentence as you recommended.

R3-6. Materials and Methods - line 106: please explain the abbreviation "HU" and add it to the text.

-> Yes, we did. We added Hounsfield Unit (HU) in lined 46.

R3-7. the "p" in p-value should be written in italics lowercase ("p"). Please change it throughout the manuscript and in the tables.

-> Yes, we did.

R3-8. Materials and Methods: please explain what the "emphysema index" is, as it is only reported in the results (line 126) without explanation.

-> We clearly mentioned “emphysema index” in the Materials and Methods.

R3-9. Table 1 and 2: please re-write the table legends, as the punctuation is incorrect: the semicolon (";") is meant to link separate items and so should be used.

-> We re-write the Table legends according to the Journal guidelines.

R3-10. Discussion: please add more recent studies to the discussion, also on the use of deep learning techniques.

-> We mentioned in the Discussion about recent researches particularly using deep learning techniques related to COPD quantification, as you recommended.

Round 2

Reviewer 2 Report

All my comments are addressed.